# NF-κB/Rel Transcription Factors in Pancreatic Cancer: Focusing on RelA, c-Rel, and RelB

**DOI:** 10.3390/cancers11070937

**Published:** 2019-07-04

**Authors:** Derya Kabacaoglu, Dietrich A. Ruess, Jiaoyu Ai, Hana Algül

**Affiliations:** 1Internal Medicine II, Klinikum rechts der Isar, Technische Universität München, 81675 Munich, Germany; 2Department of Surgery, Faculty of Medicine, Medical Center, University of Freiburg, 79106 Freiburg, Germany

**Keywords:** NF-κB, PDAC, pancreatic cancer, RelA, p65, RelB, c-Rel

## Abstract

Regulation of Nuclear factor kappa-light-chain-enhancer of activated B cells (NF-κB)/Rel transcription factors (TFs) is extremely cell-type-specific owing to their ability to act disparately in the context of cellular homeostasis driven by cellular fate and the microenvironment. This is also valid for tumor cells in which every single component shows heterogenic effects. Whereas many studies highlighted a per se oncogenic function for NF-κB/Rel TFs across cancers, recent advances in the field revealed their additional tumor-suppressive nature. Specifically, pancreatic ductal adenocarcinoma (PDAC), as one of the deadliest malignant diseases, shows aberrant canonical-noncanonical NF-κB signaling activity. Although decades of work suggest a prominent oncogenic activity of NF-κB signaling in PDAC, emerging evidence points to the opposite including anti-tumor effects. Considering the dual nature of NF-κB signaling and how it is closely linked to many other cancer related signaling pathways, it is essential to dissect the roles of individual Rel TFs in pancreatic carcinogenesis and tumor persistency and progression. Here, we discuss recent knowledge highlighting the role of Rel TFs RelA, RelB, and c-Rel in PDAC development and maintenance. Next to providing rationales for therapeutically harnessing Rel TF function in PDAC, we compile strategies currently in (pre-)clinical evaluation.

## 1. Introduction

Pancreatic ductal adenocarcinoma (PDAC), with a five-year survival rate around 9%, has a dismal prognosis [1]. Current estimates postulate an about two-fold increase in pancreatic cancer incidence by 2040 [2]. In contrast with many other solid tumors for which treatment options have shown promising improvement, PDAC still remains recalcitrant. Due to late diagnosis, surgical resection is an option for only about 10% of the patients with a localized tumor [1]; the remainder receive first-line treatment with chemotherapeutic regimens including gemcitabine, nab-paclitaxel, and FOLFIRINOX, which are selected according to the patient’s performance status [3]. Next to chemotherapeutics, only epidermal growth factor receptor (EGFR)-directed targeted therapy has proceeded to clinical application, but providing only a marginal benefit regarding patient survival [4].

For long years, PDAC has been thought to evolve from metaplastic to neoplastic stages by slowly accumulating mutations. Recent advances support a novel model in which instantaneous chromothriptic events in cells can bypass the need for this step-wise accumulation of mutations for carcinogenesis [5]. In the conventional model, PDAC forms through a sequence of driver mutations in *KRAS*, *CDKN2A*, *TP53,* and *SMAD4*, along with a highly inflammatory tumor microenvironment [6]. This model is also considered for genetically engineered mouse models (GEMMs), in which pancreas-specific expression of a mutated Kras protein (mostly Kras^G12D^) allows the progressive formation of meta- to neoplastic pancreatic lesions, mirroring the histopathological properties of the human disease [7]. The tumor microenvironment is composed of a desmoplastic stroma with an abundance of immune cells, embodying up to more than 80% of the tumor mass [8]. Studies investigating carcinogenesis in PDAC GEMMs revealed pro-tumorigenic inflammatory cell infiltration as early as in the pre-invasive neoplastic stage [9].

NF-κB signaling, as the matchmaker of cancer and inflammation [10], holds potential for cancer therapeutics, allowing a targeted strategy directed simultaneously toward cancer cells and the cancer associated inflammatory cells. The first κB DNA binding sequence motif was discovered 33 years ago by Sen [11]. Despite the complexity and unpredictability of the outcomes of NF-κB signaling, it is well accepted as a major signaling pathway connecting inflammation and cancer. Having initially evolved as a stress response pathway, NF-κB is conserved among many eukaryotic species except yeasts and *Caenorhabditis elegans*, which encode some proteins similar to NF-κB upstream signaling components but not the downstream transcription factors [12,13]. Simple organisms, like sea anemones and corals, possess transcriptionally active NF-κB transcription factors [14].

NF-κB signaling transcription factor genes are *Nfkb1* (p105-p50), *Nfkb2* (p100-p52), *Rela* (RelA, p65), *Rel* (c-Rel), and *Relb* (RelB) [15]. In an un-induced state, RelA, c-Rel, and RelB are retained in the cytoplasm through their interaction with Inhibitory κB (IκB) proteins [16,17,18,19]. p105 and p100 proteins can also act as IκB proteins due to their conserved C-terminal ankyrin repeats. However, proteolytic degradation of their C termini converts them to transcription factors (p105 to p50 and p100 to p52) [20,21,22,23,24,25,26,27]. In a simplistic overview, NF-κB signaling is divided into two, canonical and non-canonical, signaling pathways [28].

In the canonical pathway, cytokines or viral-bacterial byproducts can induce NF-κB signaling through their action on cytokine receptors, pattern recognition receptors (PRRs), and T-cell and B-cell receptors. Depending on the receptor that is activated, a series of protein cascades leads to phosphorylation-mediated activation of IκB kinase (IKK) complexes. The IKK complex is formed of three subunits: IKKα (IKK1) and IKKβ (IKK2) are the catalytic kinases, and IKKγ (NEMO) is the regulatory subunit [29,30,31,32,33,34,35]. The activated IKK complex phosphorylates downstream IκB proteins leading to their ubiquitination and proteasomal degradation. Upon IκB degradation, released RelA:p50 (the most prominent dimer), RelA:c-Re,l or c-Rel:p50 dimers translocate into the nucleus and drive target gene transcription. In a cell autonomous manner, cellular stress-associated factors (e.g., reactive oxygen species (ROS), DNA damage) can also induce the canonical NF-κB pathway in an unconventional way, starting at the intracellular level [28]. Of note, next to IκB proteins, IKKα and IKKβ may also phosphorylate RelA^S536^, with differential functional outcomes on NF-κB signaling [36,37,38,39,40,41]: IKKβ-mediated phosphorylation of RelA^S536^ is suggested to increase acetylation of RelA^K310^, associated with transactivation. Other studies in various cell systems supported this NF-κB activating function of IKKβ mediated RelA^S536^ phosphorylation. In contrast, a study with mice bearing kinase dead mutant IKKα suggested that IKKα mediated phosphorylation of RelA^S536^ in macrophages increases RelA turnover, which is functionally important for resolution of inflammatory responses [42]. In another study, phosphorylation of RelA^S536^ was proposed to not be dependent on IKKα expression, as detected in a setting with IKKα knockout embryonic macrophages [43]. Still, the influence of RelA^S536^ phosphorylation levels on RelA turnover is further corroborated with the help of a RelA^S534A^ (a non-phosphorylatable mutant version of the human RelA^S536^ homologue in murine) knock-in mouse model in which mice showed enhanced NF-κB activity [44].

Non-canonical NF-κB activation requires inducible proteolytic truncation of p100 protein to p52. Whereas processing of p105 to p50 can occur in a transcription-coupled unstimulated state, p100 processing requires the upstream action of specific ligands [45,46]. These properties reflect why the canonical pathway response is generally fast and transient, whereas the non-canonical pathway is slow and longer lasting. Some TNFR superfamily members like BAFFR [47], CD40 [48], LTβR [49] and RANK [50] induce signaling cascades activating NF-κB inducing kinase (NIK), which cooperates with IKKα to induce phosphorylation-mediated C-terminal ubiquitination of p100 via βTrCP [46,51,52]. Typically, released p52:RelB dimers translocate into the nucleus and activate downstream transcriptional targets [49,53].

In NF-κB signaling, although a variety of upstream enzymatic and scaffolding proteins connect extracellular stimuli to intracellular responses, many of these intermediate signals meet at the level of IKK proteins. Yet, considering the divergent NF-κB-dependent and -independent functions of IKKs, converging on IKK by no means has a singular impact on downstream effects [28]. Therefore, an understanding of the functions of the ultimate transcription factors RelA, c-Rel, and RelB is required. In general, NF-κB signaling refers to the activation of canonical RelA:p50 dimers. Thus, a plethora of primary literature focused on RelA in contrast to the other NF-κB components RelB and c-Rel. However, this evokes an unfavorable view, underestimating the potential functional outcomes of c-Rel and RelB containing dimers and might mislead us to consider NF-κB signaling in an overly straightforward and simplistic way.

Despite the work of almost 35 years, many questions remain to be answered about the exact mechanisms of how NF-κB functions. The gold standard research in NF-κB signaling relies on either nuclear localization of NF-κB transcription factors or on in vitro DNA binding activity (EMSA) and chromatin immunoprecipitation (ChIP) to evaluate their activity status. However, nuclear localization of NF-κB does not necessarily represent transcriptional activity; as of today, nuclear NF-κB inhibitory mechanisms are known [54,55,56,57,58,59]. Additionally, DNA binding does not exclusively imply transcriptional activation, since NF-κB may also inhibit transcription or regulate it by recruitment of co-modulator proteins [60,61,62,63,64,65].

For pancreatic cancer, which is a highly inflammatory disease, NF-κB holds promise for simultaneously targeting both cancer and cancer-associated inflammation. In both malignant and normal cells, functional outcomes of distinct signaling branches might be the same, yet it is the imbalance in overall NF-κB signaling that creates an advantage for tumor cells. It would be a mistake to associate high NF-κB activation with oncogenic potential or vice versa; it is rather a matter of the “properness” of collective signaling. Such an important and versatile signaling pathway would be expected to accumulate mutations in cancer; however, this is only true for hematological malignancies but not for many solid tumors, including PDAC [66]. Here instead, an inflammatory phenotype pressures cancer cells to evolve an addiction to NF-κB signaling. 

Currently, most of the reported research on the function of NF-κB in cancer relied on GEMMs with knockout of upstream IKK complexes. These studies revealed a complex function of NF-κB signaling in various cancer models [67,68,69,70,71,72,73]. A study of note reported the inability of a combined *Rela*, *Relb,* and *Rel* knockout to phenocopy effects seen with *Ikbkg* (NEMO) knockout in a murine hepatocarcinogenesis model, whereas the introduction of a constitutively active IKK2 is able to rescue the phenotype [74]. Considering the NF-κB-independent roles of IκBs and IKKs, a focus on the downstream NF-κB transcription factors RelA, c-Rel, and RelB is therefore essential for a clear understanding.

## 2. RelA/p65 in PDAC

Whereas the necessity of NF-κB signaling during Ras-induced transformation is still controversial, the expression of oncogenic Ras proteins is known to induce NF-κB signaling [75,76,77,78]. Constitutive canonical NF-κB signaling defined by an increased RelA nuclear localization or DNA binding activity is present in both cancer cell and histology specimens of human PDAC patients [69,79]. Therefore, a functional interaction between constitutive NF-κB signaling and *Kras* mutation in PDAC patients is predictable. One study suggested Interleukin-1α (IL-1α) as the missing link between Kras activity and constitutive NF-κB signaling [69]. Accordingly, Kras^G12D^ induces AP1 transcriptional activity and IL-1α production, which in turn stimulates NF-κB signaling. This results in a feed forward loop through the NF-κB-mediated production of more IL-1α and p62. p62 is known to regulate the ubiquitination of TRAF6 and subsequent IKK phosphorylation, triggering IκB degradation [80,81,82,83]. The interaction of p62 and TRAF6 is not required for the initial NF-κB activation but for sustained signaling. In support, PDAC patient samples revealed that IL-1α expression is positively correlated with enhanced RelA-positive nuclear staining and poor survival [69]. An inflammatory response in the Kras^G12D^ mouse model generated through sustained NF-κB signaling was proposed to further amplify the pathologic Ras activity in pancreas [84].

Phosphoinositide 3-kinase (PI3K) signaling is an effector pathway downstream of Kras. The loss of phosphatase and tensin homolog (PTEN), a negative regulator of PI3K activity, appears to cooperate with Kras signaling to further augment RelA nuclear localization and κB luciferase reporter activity in an IKK-independent manner. Whereas PI3K inhibition reduces RelA nuclear localization, PTEN knockdown has the opposite effect. Previous reports asserted a RelA-mediated regulation of PTEN expression [85]. However, this effect is not due to the classical NF-κB function of RelA. Rather, RelA sequesters and restrains p300-CREB-binding protein (CBP) transcriptional co-activator proteins. These results indicate a possible positive feedback loop between PI3K and NF-κB signaling pathways. Next to PI3K downstream of Kras, glycogen synthase kinase-3 (GSK-3) was also suggested to induce IKK dependent canonical NF-κB activity in pancreatic cancer cells [86]. Later, further insights revealed that GSK-3α stabilizes the TAK1-TAB1 complex, which is upstream of the IKK complex [87].

Redox balance is linked to NF-κB signaling on various levels. In many cell types NF-κB signaling can induce ROS scavengers and related enzymes (SOD1-superoxide dismutase 1, SOD2-superoxide dismutase 2, FHC-ferritin heavy chain, thioredoxins, glutathion S-transferases, NQO1-NAD(P)H dehydrogenase, and HO1-heme oxygenase 1) as a cellular protection mechanism; in immune cells, it is also able to induce ROS production to support phagocytosis via a number of proteins (NOX2-NADPH oxidase 2, XOR-xanthine oxidoreductase, iNOS-nitric oxide synthase inducible, COX2-cyclooxygenase 2, cytochrome p450 enzymes) [88]. Liou et al. proposed that mutant Kras induces mitochondrial metabolic stress in premalignant lesions, which in turn activates Polycystin 1 (Pkd1) and NF-κB pathways. Upon NF-κB activation, cells upregulate components of the EGFR pathway, supporting the de-differentiation of acinar cells [89]. Pkd1 was also stated to induce acinar cell reprogramming in a Notch-dependent manner, implying a possible convergence with NF-κB signaling in pancreatic carcinogenesis [90].

Constitutive RelA transcription factor activity in tumors is mainly associated with an inflammatory cytokine network inducing NF-κB signaling [69,91]. However, this induction requires an activated IKK complex, which is not observed consistently throughout many cancer samples, implying the presence of other downstream mechanisms to prolong RelA transcriptional activity. Phosphorylation of Stat3 and its oncogenic activity was demonstrated to be constitutive due to both cell intrinsic and tumor microenvironment (TME) cross talk in PDAC [92,93,94,95]. Nuclear phospho-Stat3 prolongs RelA nuclear localization via increasing its acetylation in various cancer cell lines [96]. Mechanistically, phospho-Stat3 recruits acetyltransferase p300 to RelA. However, deletion of Stat3 in the myeloid compartment also reduces RelA acetylation in tumor cells, signifying the importance of the initial cytokine network to maintain constitutive RelA activity [96]. Importantly, this connection still needs to be proven for PDAC.

The interaction between p53 and RelA signaling is mostly related to their impacts on tumor metabolism. Previous reports indicated a duality for the relevance of p53 status on RelA activity in different Ras-driven lung tumorigenesis mouse models [77,78]. Studies focusing on mouse embryonic fibroblasts (MEF) cells proposed that RelA can directly increase p53 transcription for metabolic adaptation to glucose starvation [97]. This increase in p53 induces a reversal of the Warburg effect, with cells shifting their metabolism from aerobic glycolysis to oxidative phosphorylation (OXPHOS) to supply their ATP demand. These results were validated in a transplantation model with human colorectal cancer cells in which RelA knockdown sensitized cancer cells to metformin (reduces systemic glucose availability and OXPHOS) and induced cell death. Loss of p53 in MEF cells was asserted to activate RelA through an increase in IKK kinase activity, resulting in a Warburg effect phenotype [98,99]. p53-RelA metabolic crosstalk is also evident in mitochondria. In contrast to its regular function in the nucleus, RelA was claimed to inhibit mitochondrial DNA (mtDNA) transcription in mitochondria, reducing the production of the proteins of the respiratory chain, relevant for OXPHOS [100]. A direct response of RelA to Tumor necrosis factor-α (TNFα) and TNF-related apoptosis inducing ligand (TRAIL) in mitochondria may accelerate the metabolic responses to external stimuli, avoiding nucleus-to-mitochondria signaling [101,102]. Whereas RelA does not contain a mitochondrial targeting sequence (MTS), its shuttling is facilitated by Mortalin (mtHSP70). p53 can inhibit the RelA-Mortalin interaction [100]. Although the metabolically relevant interactions between p53 and NF-κB signaling are well accepted, they have not been elucidated or confirmed in pancreatic cancer.

A dual function of RelA in PDAC carcinogenesis and persistency was revealed with the use of a mouse model in which the nuclear localization signal (NLS) of RelA is conditionally deleted [103,104]. Unlike malignant counterparts, normal cells generally undergo senescence in response to various stress inducers [105]. *Rela* knockout MEF cells are able to bypass senescence, leading to earlier immortalization compared with *Rela* WT cells. Due to impaired DNA damage repair signaling, *Rela* knockout MEF cells are prone to accumulating genomic rearrangements, facilitating their immortalization [106]. In support of this, RelA truncation accelerates the carcinogenesis in Kras^G12D^ mediated pancreatic carcinogenesis. Oncogene-induced senescence (OIS) is an important barrier in pancreatic carcinogenesis as most of the premalignant lesions are stuck in this stage [107,108]. Senescent cells secrete various cytokines, chemokines, proteases, and growth factors in order to create a network with the neighboring cells, namely senescence associated secretory phenotype (SASP) [109,110,111]. As observed in MEF cells, SASP induction by RelA is also evident in pancreatic premalignant lesions [104,112]. The tumor suppressor function of RelA was suggested to convert to an oncogenic function in mouse models with concomitant *Trp53* or *Ink4a/Arf* deletion, in which the senescence barrier is exceeded [113,114,115]. Murine Cxcl1 (and its human functional homologue IL-8) was identified as a major SASP component, which signals through CXCR2 in an autocrine manner to sustain the senescent phenotype in pancreatic premalignant lesions [104,116].

In a recent study, Jin et al. uncovered the mechanism through which tumor suppressor retinoblastoma (RB) protein in a hyperphosphorylated state may diminish RelA-induced PD-L1 production [117]. The study has important implications for immunotherapy, as the ectopic expression of an RB phosphomimetic is able to inhibit cancer cell PD-L1 production in various entities, including PDAC. Therapeutic exploitation of this interaction might provide significant benefits to convert generally immunologically “cold” pancreatic tumors to “hot” ones, making them susceptible to immunotherapy [118].

DNA damage repair proteins meet with RelA at crossroads in chemotherapy response. DNA damage-induced Ataxia telangiectasia and Rad3-related (ATR) protein is shown to activate Checkpoint kinase 1 (Chk1), which phosphorylates RelA^T505^ [119,120]. As a result, Claspin expression maintains Chk1 activity, implicating a feed-forward loop. This mechanism indicates another suppressive function of RelA in tumorigenesis, in which a RelA-mediated cell cycle checkpoint prevents exacerbation of genomic instability [121,122]. In established cancer, different chemotherapeutic agents variably affect RelA activity depending on the drug used and its mode of action [123,124,125]. Yet, for PDAC, a number of NF-κB/RelA-dependent mechanisms of chemotherapy resistance have been proposed. RelA RNA-interference (RNAi) was shown to synergize with gemcitabine in pancreatic cancer cells [126]. Anakinra (IL-1R inhibitor) also synergizes with gemcitabine in human cell orthotopic transplantation models through a decrease in RelA activity [127]. Then, the gemcitabine transporter hCNT1 was demonstrated to be negatively regulated by MUC4 through RelA:p50 NF-κB signaling [128]. Pancreatic cancer stem cells, which are resistant to gemcitabine treatment, were also proposed to maintain their stemness at least partially through NF-κB signaling [129]. Additionally, a transcriptomics analysis performed on cisplatin-resistant human pancreatic cell lines revealed a dysregulation of NF-κB signaling [130]. 

A concise schematic for the here-collated relevance of RelA signaling and its crosstalk with other pathways is depicted in Figure 1. Manipulation of NF-κB signaling at the level of IKK complexes inevitably alters activities of parallel pathways like Notch, Klf-related, and p62-TRAF6 [69,72,73]. Affirming this, a transcriptomics analysis comparing the *Ikbkb* deletion and RelA truncation murine PDAC models revealed profound differences in the profiles of enriched pathways [104]. Considering cumulative evidence, therapeutically targeting RelA signaling requires a renewed approach based on a deeper understanding of its function.

## 3. c-Rel in PDAC

c-Rel, considered a canonical NF-κB signaling transcription factor, was identified as a homolog of v-Rel, an avian reticuloendotheliosis virus strain T protein [131,132]. An oncogenic transformation assay revealed that in contrast to the other NF-κB transcription factors, only mouse and human c-Rel had the ability to transform chicken primary spleen cells [133]. In congruence, a transgenic mouse model with c-Rel overexpression driven by a *MMTV-LTR* promoter is able to form mammary tumors with secondary driver events [134,135]. Structurally, c-Rel shows similarities with RelA and RelB having Rel homology domain (RHD) and transactivation domain (TAD) domains, though the target DNA binding preference can show variance [136,137]. Despite c-Rel being attributed mostly to hematological malignancies, growing evidence suggests important functions in solid carcinomas [136,138].

Ras-mediated transformation of MEF cells requires neither RelA nor c-Rel, but both enhance it [75]. In support of this, c-Rel nuclear localization is enhanced in the aforementioned Ras transformed mouse lung tumor model [77]. An RNAi screen identified TANK Binding Kinase 1 (TBK1) as a top candidate synthetic lethality partner of mutant Kras in various human cancer cell lines [139]. Mutant Kras signals through many downstream effectors, like PI3K, Raf kinases, and RalGEFs. Unlike RalB, depletion of Raf and PI3K does not result in synthetic lethality with the Kras mutation, supporting previous reports in which RalB-SEC5 was identified as a TBK1 activator [139,140]. The TBK1-IKKε complex is also known to phosphorylate c-Rel, enhancing its nuclear localization in HEK 293T cells [141]. In support, a lower analysis threshold also revealed c-Rel as a candidate synthetic lethality partner for mutant Kras, next to TBK1. Mechanistically, the mutant Kras-RalB-TBK1 axis propagates c-Rel transcriptional activity to induce Bcl-xL (an anti-apoptotic protein) whose overexpression rescued synthetic lethality following suppression of TBK1 [139]. Whereas this study excluded pancreatic cancer cell lines, additional work confirmed the importance of the same signaling axis in the erlotinib (EGFR inhibitor) resistance of this entity [142]. Here, it was suggested that erlotinib-resistant cancer cells develop a stemness-like phenotype through recruitment of a Kras-RalB complex by α5β3 integrin. Analogously, this axis induces an NF-κB signaling cascade involving c-Rel activation. A proteasome inhibitor bortezomib (FDA-U.S. Food and Drug Administration approved), which is known to inhibit IκB degradation, diminished both intrinsic and acquired erlotinib resistance and reduced tumor stemness [142]. Yet, the response of pancreatic cancer cells acquiring an active Kras signature to TBK1 inhibition remains controversial [143,144].

The aforementioned DNA damage-induced ATR-Chk1-NF-κB-Claspin signaling axis also involves c-Rel for modulation of cell cycle arrest [122,145]. The Cancer Genome Atlas (TCGA) data revealed a negative correlation between *Clspn* expression and prognosis in pancreatic cancer patients. This may further emphasize the possible dual, converse role of both c-Rel and RelA (as mentioned above) during carcinogenesis versus in established cancer. A low Claspin amount is beneficial for carcinogenesis in order to create genomic instability to pass the Hayflick limit, but an excess genomic instability may become detrimental for formed cancer cells and predispose to vulnerability toward certain chemotherapeutics. Despite no study being published yet regarding the function of c-Rel in pancreatic carcinogenesis, the RelA truncation mouse model strikes with more DNA damage induced γ-H2AX in premalignant lesions, accompanied by accelerated carcinogenesis [104]. An illustrated overview of c-Rel-dependent facets in (pancreatic) cancer is provided in Figure 2.

Regulatory T lymphocytes (T_reg_) are known for their ability to limit antitumor immunity in the tumor microenvironment, and targeting their action may enhance immunotherapy response of tumors [118]. c-Rel is known to be an important transcription factor for the thymic development of T_reg_ cells [146,147,148], whereas RelA was proposed to be important in maintaining T_reg_ identity [149]. Comparing the embryonic lethality of a global *Rela* knockout with the comparably mild phenotype of a dysregulated humoral immune system in global *Rel* knockout mice, targeting c-Rel would potentially be rather fruitful with less systemic adverse effects [150,151]. Inhibition of c-Rel with either pentoxifylline or IT-603 enhanced growth inhibition of tumors by anti-PD-1 immunotherapy in a melanoma transplantation mouse model [152,153,154]. In line with this, T_reg_ depletion improved antitumor CD8^+^ cytotoxic T cell recruitment in orthotopic PDAC transplantation models, whereas in the endogenous genetic PDAC model, this effect was not evident [155,156]. Therefore, a combination of T_reg_ depletion with immune checkpoint inhibition might be of value in PDAC as well, but has not been investigated yet. Targeting c-Rel in this manner might demonstrate an anti-tumor effect both on the cancer cell and immune system levels to convert an immunologically “cold” tumor into a “hot” one, thereby enhancing immunotherapy efficiency [118]. 

## 4. RelB in PDAC

The RelB:p52 heterodimer is the general, transcriptionally active downstream component of the non-canonical NF-κB signaling pathway. RelB can also form heterodimers with RelA and p50. Although RelA:RelB heterodimers are known to be transcriptionally inactive, RelB:p50 is formed in a p100 processing-dependent manner upon selective ligand activation [157,158,159,160,161]. The canonical and non-canonical signaling pathway share transcriptional targets, yet nucleotide variations in κB DNA binding sequences or distinct conformations of the dimers with recruitment of various co-activator/repressor proteins can diversify the target preference [162].

Constitutive activation of non-canonical NF-κB, along with NIK stabilization and constitutive p100 phosphorylation has been detected in human PDAC cell lines [163]. In human PDAC tissues, NIK activity is correlated with TRAF2 downregulation, especially in moderately or poorly differentiated subtypes [164]. Other than NIK stabilization, GSK-3α also regulates the nuclear p52 amount, affecting the non-canonical signaling pathway activation in human PDAC lines [87]. In murine models, the *Nfkb2* gene was identified as a proto-oncogene, along with *Myc* and *Yap1*, and its genetic amplification appears to be sufficient to drive PDAC without the need for *Kras*^G12D^ dosage gain [165].

A direct oncogenic function of RelB activity in pancreatic carcinogenesis was revealed with the help of a *Relb* knockout mouse model. Loss of RelB resulted in a decreased amount of pre-neoplastic structures and delayed carcinogenesis. Mechanistically, both in Kras^G12D^ premalignant lesions and established PDAC cells, nutrient deprivation activates Nuclear Protein 1 (Nupr1), inducing RelB expression, which in turn protects from apoptosis through transcription of immediate early response 3 (*Ier3*). These results enabled the hypothesis of a general oncogenic activity of RelB in both carcinogenesis and tumor maintenance [166]. Nupr1 also negatively regulates senescence in Kras^G12D^-induced pancreatic and lung carcinogenesis [167]. Although the role of RelB in this axis is being questioned, this may imply opposite functions of RelA and RelB in pancreatic carcinogenesis through regulation of senescence [104,167]. Next to anti-apoptotic gene induction, RelB:p52 heterodimers regulate the G1 to S cell cycle progression through the S-phase kinase associated protein 2 (Skp2) and p27^Kip1^ axis in human PDAC cells [168].

Enhancer of zeste homolog 2 (EZH2) is a polycomb repressive complex 2 (PRC2) component catalyzing the tri-methylation of H3K27 residues to repress gene transcription [169]. Although specific evidence for PDAC is lacking, non-canonical NF-κB induces EZH2 which mediates a bypass of senescence by repressing p53/Rb, p16^Ink4a^, and p14^ARF^ [170,171,172]. In pancreatic cancer, EZH2 negatively regulates p27^Kip1^ expression in a RelB:p52-dependent manner [173]. Other than its transcriptionally repressive role within PRC2, EZH2 may also exert transcription-activating functions in a non-canonical fashion. In cancer cells with mutations in the SWItch/Sucrose non-Fermentable (SWI/SNF) complex, this non-canonical function of EZH2 is necessary, although this necessity can be bypassed by introduction of a Ras mutation in various cancer cells [174]. This EZH2 dependency might generate a vulnerability in 30%–40% of pancreatic cancer patients whose tumors harbor mutations in the SWI/SNF complex [175,176,177]. However, treatment with an EZH2 catalytic activity inhibitor did not affect cyst development in the *Arid1a* knockout (a SWI/SNF complex compound) Kras^G12D^ pancreatic mouse model, even if reduced histone methylation was observed [178]. This indicates the need for alternative EZH2 inhibitors blocking its non-canonical functions. The contribution of RelB specifically with respect to the non-canonical function of EZH2 is still unknown; however, considering the transcriptional induction of EZH2 by RelB, targeting non-canonical NF-κB may be of therapeutic value in SWI/SNF mutant PDAC patients. A schematic presenting the role of RelB in pancreatic cancer is provided in Figure 3.

Cell death evoked by radiotherapy may have immunogenic effects, triggering the recruitment and activation of anti-tumoral immune cells, including dendritic cells (DCs) and cytotoxic T cells [118]. Those activated immune cells can in turn secrete type I interferons to augment cytotoxicity and DC antigen presentation [179,180]. Whereas the canonical NF-κB pathway drives interferon production, the non-canonical pathway inhibits it. Thus, application of canonical NF-κB inhibitors was reported to impede radiotherapy response, whereas non-canonical inhibitors augmented it [181]. These results point to a complex but anticipated differential function of NF-κB downstream transcription factors in cancer therapeutics, again signifying their differences. 

## 5. Conclusions

A generic oncogenic impression of NF-κB signaling requires a detailed map of the spatiotemporal mode of action for each individual component. The oncogenic or tumor suppressor functions of RelA, c-Rel, and RelB can be highly context-specific, showing variation not only in different cell types but also depending on the stage of malignant transformation. Despite the abundance of NF-κB inhibitors, their efficacy in clinical use is controversial [182]. A collection of selected clinical trials targeting NF-κB signaling or its associated pathways as mentioned in this review is listed in Table 1. Unfortunately, no compounds haveyet been found to specifically target RelA, c-Rel, or RelB in clinical evaluation. Besides, NF-κB signaling is complex and considered to be fundamental not only for cancer cells but also for stromal cells. Therefore, even more specific inhibitors not targeting individual NF-κB transcription factors as a whole but rather their distinct functions may hold promise for cancer therapeutics.

## Figures and Tables

**Figure 1 cancers-11-00937-f001:**
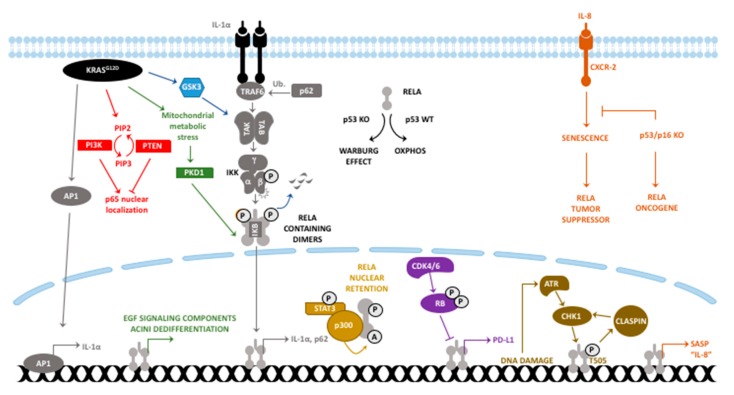
Reported and potential oncogenic and tumor suppressor functions of RelA in pancreatic carcinogenesis and tumor maintenance are depicted. Each color indicates an individual signaling axis although crosstalk is highly expected. P: Phosphorylation, A: Acetylation, Ub: Ubiquitination.

**Figure 2 cancers-11-00937-f002:**
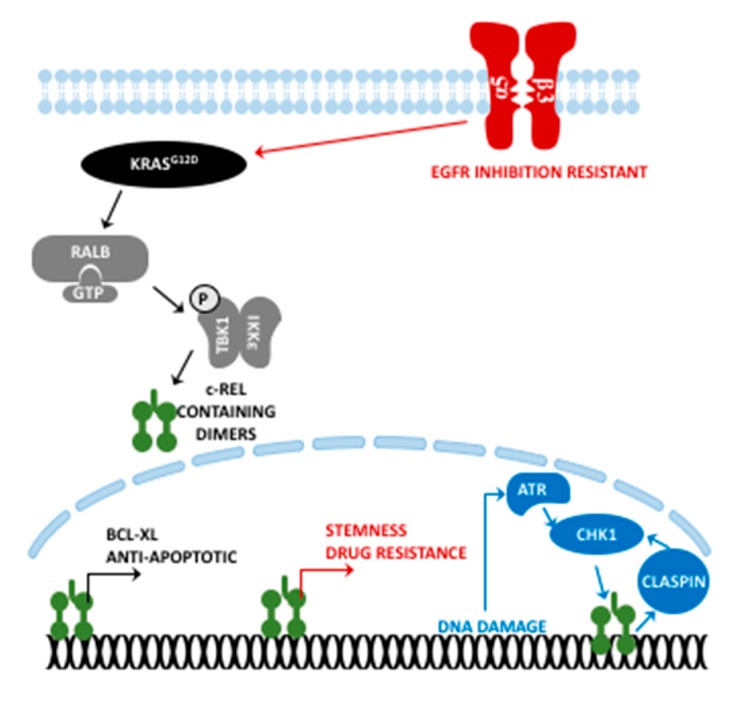
Functions of c-Rel in pancreatic cancer. Although some of these pathways still need to be confirmed in PDAC, similarities are expected. Each color represents a different signaling axis. P: Phosphorylation.

**Figure 3 cancers-11-00937-f003:**
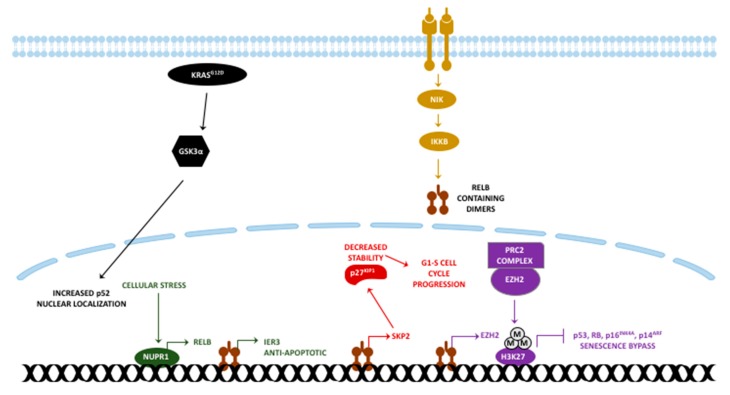
Functions of RelB in pancreatic cancer. Although the EZH2 connection still needs to be confirmed for PDAC, similarities are expected. Each arrow with a different color represents a different signaling axis. M: Methylation.

**Table 1 cancers-11-00937-t001:** A selection of clinical trials for pancreatic cancer targeting NF-κB signaling or its associated pathways are given. Abbreviations as followed; n.a.: non-applicable, HDACi: histone deacetylase inhibitor, NK: natural killer cells, IL-1R: interleukin 1 receptor.

NCT Number	Intervention	Disease	Phase	Status	Ref.
NCT01056601	Bortezomib (Proteasome/NF-κB Inhibitor) + Panobinostat (HDACi)	Pancreatic cancer progressive upon gemcitabine treatment	II	Terminated: Toxicity and lack of response.	[183]
NCT00416793	Bortezomib (Proteasome/NF-κB inhibitor) + Carboplatin (chemotherapy)	Metastatic pancreatic cancer	II	Terminated: Toxicity and lack of response	n.a.
NCT00052689	Bortezomib (Proteasome/NF-κB inhibitor) ± Gemcitabine (chemotherapy)	Stage IV pancreatic cancer	II	Completed: Results n.a.	n.a.
NCT00622674	Bortezomib (Proteasome/NF-κB inhibitor) + Cetuximab (EGFR inhibitor)	EGFR-expressing solid tumors; 3/37 patients with pancreatic tumor	I	Completed: Treatment dose is tolerable; no response observed in pancreatic cancer	[184]
NCT00052689	Bortezomib (Proteasome/NF-κB inhibitor) ± Gemcitabine (chemotherapy)	Older patients with advanced pancreatic cancer	II	Completed: Results n.a.	n.a.
n.a.	Bortezomib (Proteasome/NF-κB inhibitor) + Paclitaxel (chemotherapy)	Advanced solid tumors	I	Completed: Manageable toxicity profile; 7/45 patients showed disease stabilization; 3 had metastatic pancreatic cancer	[185]
NCT03878524	Various targeted/chemotherapy drugs in combination, among them Bortezomib (Proteasome/NF-κB inhibitor)	Advanced cancers including pancreatic	I	Not yet recruiting. The study aims at molecular stratification and combination treatment in a personalized approach.	n.a.
NCT00720785	Autologous, ex vivo expanded NK cells ± Bortezomib (Proteasome/NF-κB inhibitor)	Various cancers including metastatic pancreatic adenocarcinoma	I	Recruiting	n.a.
NCT00094445	Curcumin (Pleiotropic signaling modulator/NF-κB inhibitor)	Advanced pancreatic cancer; no concomitant chemo/radiotherapy	II	Completed: no toxicity; 2/21 patients showed biological activity.	[186]
NCT00192842	Curcumin (Pleiotropic signaling modulator/NF-κB inhibitor) + Gemcitabine (chemotherapy)	Advanced pancreatic cancer	II	Completed: low compliance for high dose oral curcumin in combination with gemcitabine; 1/11 with partial response, 4/11 with stable disease	[187]
NCT02336087	Gemcitabine + nab-Paclitaxel (Chemotherapy) + Metformin + Dietary supplement including curcumin	Unresectable pancreatic cancer	I	Recruiting	n.a.
NCT03382340	IMX-110: nanoparticle encapsulating curcumin and low-dose doxorubicin (chemotherapy)	Advanced solid tumors	I/II	Recruiting	n.a.
NCT02671890	Gemcitabine (chemotherapy) ± Disulfiram (Proteasome/NF-κB inhibitor)	Unresectable solid tumors or metastatic pancreatic cancer	I	Recruiting	n.a.
NCT02550327	Gemcitabine, Nab-Paclitaxel, Cisplatin and Anakinra (IL-1R antagonist)	Localized pancreatic ductal adenocarcinoma	Early I	Completed: Combination is tolerable. Further analysis pending	[188]
NCT01632306	LY2090314 (GSK-3 inhibitor) in combination with various chemotherapy regimens (Gemcitabine, FOLFOX, nab-Paclitaxel)	Metastatic pancreatic cancer	I/II	Terminated: Lack of patient enrollment.	n.a.
NCT03678883	9-ING-41 (GSK-3β inhibitor) ± various chemotherapy regimens	Advanced cancers including pancreatic	I/II	Recruiting	n.a.
NCT03454035	Palbociclib (CDK4/6 inhibitor) Ulixertinib (ERK1/2 inhibitor)	Advanced pancreatic cancer and other solid tumors	I	Recruiting	n.a.
NCT02465060	Various targeted therapies including Palbociclib (CDK4/6 inhibitor) BAY 80-6946 (PI3K inhibitor)	Multiple tumor types including refractory pancreatic cancer, treatment option to be evaluated based on genetic testing	II	Recruiting	n.a.
NCT03065062	Palbociclib (CDK4/6 inhibitor) Gedatolisib (PI3K/mTOR inhibitor)	Various solid tumors including advanced pancreatic cancer	I	Recruiting	n.a.
NCT03682289	AZD6738 (ATR inhibitor) ± Olaparib (PARP inhibitor)	Various solid tumors including advanced pancreatic cancer	II	Recruiting	n.a.

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
