# Peer review of "NF-κB/Rel Transcription Factors in Pancreatic Cancer: Focusing on RelA, c-Rel, and RelB"

_cancers, 2019, doi:10.3390/cancers11070937_

Round 1
Reviewer 1 Report
Kabacaoglu et al.; Cancers 2019
General comments:
In the introduction part describing the NF-κB system, the authors by and large reiterate many previous reviews, without really providing a critical survey and most recent advances of the field. Btw: NF-κB, not NFκB, is the preferred notation.
The review then focusses in depth on the literature describing the role of RelA, c-Rel and RelB in pancreatic ductal adenocarcinoma, the field of notable expertise of the senior author and his group. The PDAC literature is well covered and up to date, with >50% of papers published within the past decade.
In general, authors could at some places better differentiate solid, reproduced and generally accepted findings or conclusions (cited as facts) from suggested findings of single studies (cited as “suggested”, “reported”, or “claimed” etc).
Otherwise, this appears to be a nice current overview of this special field.
Some specific points to be addressed:
Line 67: is formed by three subunits, not kinases
Line 69: IKK also phosphorylates p65, for example, which is functionally important and should be mentioned
Line 70: c-Rel:p50 dimers rapidly translocate, who showed this really?
Line 80: Probably reference 37 is wrong here.
Line 87: what is meant by “very downstream”
Line 198: Cxcl1/IL8 should be Cxcl8/IL8
Line 492: Reference 51, incomplete (clin cancer res)
Line 596: Reference incomplete
Author Response
We really appreciate the reviewer for her/his time and valuable comments. Based on the highlighted problems, we made the requested changes.
1) The aim of the review is rather to provide an overview on the data available about RelA, c-Rel and RelB in pancreatic cancer. By doing this we aimed to set the platform for a comprehensive and critical analysis. This was provided in every section of the review. Due to the overwhelming data on NF-kB we wished to focus on the discussion centered around this signaling pathway and pancreatic cancer. The inclusion of the data on complexity of NF-kB signaling and its effect on other cancers would exceed the scope of the manuscript and would confuse the overall message.
2) As requested, we changed all NFkappaB to NF-kappaB.
3) We appreciate the comment of the author regarding the use of the proper word while stating whether the data it is solid or not yet reproduced. As suggested, on certain sentences we made changes.
4) "kinases" is changed with "subunits".
5) We wrote an additional paragraph regarding the IKK mediated phosphorylation of RelA. Of course this could rather be explained in a longer manner, however since our focus is not to explain NF-kB signaling in detail, we kept it rather brief.
6) As reviewer highlights, it is not really shown that c-Rel:p50 dimers rapidly translocate to nucleus, at least not as fast as p65 dimers. Since this topic would also require a large explanation, we decided to delete only "rapidly" part.
9) The reference is replaced with the proper ones. We specifically thank to you for the suggestion.
10) By "very downstream" we meant the transcriptions factors which are showing the final actions in a signaling pathway. However, we realized that it may cause confusion. Therefore, we removed the "very".
10) Cxcl1/IL8 writing aimed to indicate mouse vs human homologs. We made the proper changes to clear it out.
11) The highlighted references with the formatting problems are fixed.
Also, in general we fixed our problems in grammar and format to enhance the quality.
With these modification we hope that the reviewer feels that this review is a valuable contribution to the field of NF-kB and its complex roles in pancreatic cancer.
Reviewer 2 Report
The manuscript entitled "Rel Transcription factors in pancreatic cancer: Focusing on RelA, c-Rel and RelB" by Derya Kabacaoglu, Dietrich A. Ruess, Jiaoyu Ai, and Hana Algül describes different roles of RelA, c-Rel, and RelB transcription factors in pancreatic cancer and PDAC treatment. The Authors show the importance of both canonical and non-canonical NF-κB signaling and other signaling pathways related to NF-κB in PDAC development.
The manuscript is well-structured and discussed, however, there are certain improvements needed:
1. Please provide a separate table with clinical trial status on therapeutic interventions in PDAC.
2. Please provide proper citations (lines 81, 101, 164, 243).
3. Line 54: yeast --> yeasts, C. Elegans --> C. elegans
4. Line 57: "In an un-induced state, these proteins..." Please specify: RelA, RelB, and c-Rel".
5. Line 66: inhibitory kappa B kinase --> IκB kinase
6. Please unify protein names (e.q., RelA not Rel-A; NF-κB not NFkB, TRAF6 not Traf6, etc.).
7. Please check the entire manuscript for proper nomenclature and abbreviations.
Overall, this manuscript can be considered for publication after necessary changes are introduced.
Author Response
We appreciate the comments of the reviewer as they carry very valuable points for the improvement of the manuscript. Based on the recommendations, we made the changes as given below:
1) We generated a clinical trial table with some selected studies. Since the focus of the review is rather downstream transcription factors and there are no specific inhibitors of them in clinic, we on purpose avoided to generate a table in the first draft. However, we believe the table will still be valuable for the reader.
2) The relevant citations are added to the recommended sentences.
3-4-5) The changes are done as requested.
6-7) For protein and gene names, in our first draft we aimed to follow the general rule for mouse vs human nomenclature. Basically, depending on the organism the study is made, we tried to write the names in format. However, based on the reviewer's suggestion, we agreed to have a uniformity in the nomenclature within the manuscript for an ease of reading.
Also, in general we fixed our problems in grammar and format to enhance the quality.
With these modification we hope that the reviewer feels that this review is a valuable contribution to the field of NF-kB and its complex roles in pancreatic cancer.
Reviewer 3 Report
The manuscript by Dr. Derya Kabacaoglu and colleges provides a balanced and comprehensive review on the roles of Rel TFs in pancreatic cancer, which is a timely and interesting topic for the readership of Cancers. The manuscript is well written, and ready to be accepted for publication.
Author Response
We really appreciate the kind comments of the reviewer about the manuscript. Although, based ont he comments of the other reviewers we made some changes. We hope that he/she is also content with these alterations.
Thank you
Round 2
Reviewer 2 Report
The manuscript entitled "Rel Transcription factors in pancreatic cancer: Focusing on RelA, c-Rel and RelB" by Derya Kabacaoglu, Dietrich A. Ruess, Jiaoyu Ai, and Hana Algül describes different roles of RelA, c-Rel, and RelB transcription factors in pancreatic cancer and PDAC treatment. The Authors show the importance of both canonical and non-canonical NF-κB signaling and other signaling pathways related to NF-κB in PDAC development.
The manuscript is well-structured and discussed.
There are few additional corrections in the text needed:
Line 64: yeasts ---> yeasts (please delete italic font)
Line 334: IKB ---> IκB
Line 374: PDAC: ---> PDAC (please delete ":")
Line 431: dendritic cell --> DC
Overall, the manuscript deserves publication after minor text correction is made.